# Natural Semantic Networks of the Neurorehabilitation Concept by Spanish Physiotherapists—A Qualitative Phenomenological Representational Study

**DOI:** 10.3390/bs13120972

**Published:** 2023-11-26

**Authors:** Javiera Andrea Ortega-Bastidas, Patricia Martín-Casas, Susana Collado-Vázquez, Cecilia Estrada-Barranco, Ismael Sanz-Esteban, Mónica Yamile Pinzón-Bernal, Paulina Ortega-Bastidas, Roberto Cano-de-la-Cuerda

**Affiliations:** 1Medical Education Department, Faculty of Medicine, Concepción University, Chacabuco Esquina Janequeo s/n., Concepción 4030000, Chile; javieraortega@udec.cl; 2Radiology, Rehabilitation and Physiotherapy Department, Faculty of Nursing, Physiotherapy and Podiatry, Universidad Complutense de Madrid, San Carlos Clinical Hospital Health Research Institute, Av. Séneca, 2, 28040 Madrid, Spain; pmcasas@enf.ucm.es; 3Physical Therapy, Occupational Therapy, Rehabilitation and Physical Medicine Department, Faculty of Health Sciences Department, King Juan Carlos University, Av. de Atenas, s/n., 28922 Madrid, Spain; roberto.cano@urjc.es; 4Physiotherapy Department, Faculty of Health Sciences, Universidad Europea de Madrid, C. Tajo, s/n, 28670 Madrid, Spain; cecilia.estrada@universidadeuropea.es (C.E.-B.); ismael.sanz@universidadeuropea.es (I.S.-E.); 5Human Movement Department, Universidad Autónoma de Manizales, Antigua Estación del Ferrocarril, Manizales 170001, Colombia; myamile@autonoma.edu.co; 6Kinesiology Department, Faculty of Medicine, Universidad de Concepción, Chacabuco Esquina Janequeo s/n., Concepción 4030000, Chile

**Keywords:** physiotherapists, neurorehabilitation, natural semantic networks, neurological rehabilitation

## Abstract

The Natural Semantic Networks (NSN) model is highly useful in analyzing the words that define a concept in terms of the value, strength, weight, or density that a specific population assigns to the construction of a learned concept. The main objective of this study was to describe the conceptualization of the concept of neurorehabilitation by Spanish physiotherapists specializing in this field using NSN. A phenomenological study is presented. The participants were physiotherapy professionals who graduated from three Spanish universities and were working in the field of neurorehabilitation. A questionnaire was administered via Google Forms, which was constructed using the NSN technique. A total of 191 physiotherapists participated in this study. The Spanish physiotherapists interviewed used a total of 1247 defining words for the concept of neurorehabilitation. The semantic core of the concept was mainly formed by the words ‘treatment’, ‘recovery’, ‘functionality’, ‘neuroplasticity’, and ‘learning’, which carried significant weight. Results were also presented taking into account the academic level and years of professional experience of the sample. The semantic network observed in this study allows us to elucidate the polysemy of the concept of neurorehabilitation, which is composed not only of certain associated words but also the meanings they imply.

## 1. Introduction

The term neurorehabilitation is a neurological compound derived from two existing components (‘neuro’ and ‘rehabilitation’) that have their origins in Greek and Latin [1]. The first component, ‘neuro’ is taken as a prefix for ‘nerve-nervous system’ and the second component, ‘rehabilitation’ is derived from the verb ‘to rehabilitate’ from Latin ‘rehabilitare’, modified proverbially with the verbal prefix ‘re’ (meaning ‘back’) and from Medieval Latin ‘habilitare’ (meaning ‘to make able’) [2].

Rehabilitation was defined in 2006 by the World Health Organization (WHO) as “an active process through which individuals with disabilities resulting from illness or injury achieve full recovery, or if full recovery is not possible, develop their maximum physical, mental, and social potential and are integrated into the most appropriate environment” [3]. In 2021, the definition was updated to indicate that “rehabilitation is a set of interventions designed to optimize functioning and reduce disability in individuals with health conditions interacting with their environment”. Therefore, phenomena related to the concept of rehabilitation include the diagnosis, evaluation, prevention, and treatment of impairments in structures and functions, activity limitations, and participation restrictions. In this context, the epidemiological justification of neurological disorders highlights that rehabilitation will be key in the therapeutic management of individuals in such circumstances and, consequently, in their functioning and disability [4].

Due to the previous lack of knowledge about the plasticity of the human nervous system, the scientific origins of neurorehabilitation are relatively recent. The term neurorehabilitation is understood as the process aimed at reducing the impairment, activity limitation, and participation restriction experienced by individuals because of a neurological disease. The professionals involved in this field aim to reduce the degree of functional impairment in patients. It should be understood as an educational and dynamic process based on the adaptation of the individual and their environment to neurological deterioration [5]. The WHO itself has defined it as “an active process through which disabled individuals due to neurological injury or disease achieve complete recovery or, if not possible, can optimize their physical, mental, and social potential and integrate into the most appropriate environment” [3].

However, the concept of neurorehabilitation is relatively recent [3] and is not without conceptual differences among healthcare professionals. Therefore, it is of utmost importance to understand the perceptions of these professionals regarding its conceptualization. One way to access the meanings that professionals assign to a concept or term is through semantic memory, which allows us to investigate how a specific population is conceptualizing, interpreting, or employing, in this case, the term neurorehabilitation. Semantic memory is acquired through specific episodes and is necessary for language use, as it enables us to organize the knowledge a person possesses about words and other verbal symbols, their meanings and referents, and the relationships between them, as well as the rules, formulas, and algorithms for manipulating these symbols, concepts, and relationships [3].

Several authors have described different physiological approaches for measuring psychological meaning as semantic generalization experiments, techniques of free associations and semantic differential. However, of the aforementioned methods which emerged in the middle of the 20th century, none had been able to fully explain psychological meaning [6,7].

From cognitive psychology, various models have been developed that enable qualitative analysis of semantics [8], among which the Natural Semantic Networks (NSN) model stands out. This model has been used in different contexts, including medicine [9], to observe how people perceive or conceptualize different aspects of their academic, political, or social environment for exploratory, descriptive, or action-promoting purposes [9]. Therefore, the NSN model is highly useful in analyzing the words that define a concept in terms of their value, strength, weight, or density in the construction of semantic networks around the learned concept [8]. Through this process, it is possible to understand what healthcare professionals understand by the term ‘neurorehabilitation’ through NSN. It might be interesting to have a better knowledge about how healthcare professionals define the term itself and also to know what other terms they do think are related to the neurorehabilitation concept, considering the academic level and years of professional experience.

Semantic memory transforms into networks or interconnections of nodes (concepts, facts, or actions that the individual considers important and connects and relates to each other, forming maps or networks) that represent the individual’s level of knowledge [10], and through various types of links in the form of networks of words and events, the meanings that the concept would present for everyone [10]. This model offers the possibility of understanding and strengthening (or modifying) the structures and processes of thought through the analysis of its results. Therefore, this analysis is pertinent for the aim of this study, which is a descriptive exploratory strategy on how the concept of neurorehabilitation is being conceptualized. There are other types of discursive strategies that could have been used but that do not correspond to the level of analysis and the scope of this type of study. The NSN technique is easily applicable and it presents a dynamic nature that interacts with other memory processes [9]. Moreover, it is one of the easily applicable alternatives to evaluate complex aspects of human thinking [11].

The semantic structure does not remain immutable, but rather develops and, therefore, more relationships are acquired as the individual’s general knowledge increases; in addition, it is also modified according to the influence of the surrounding culture and their own life experience [7].

Thus, cognitive structures such as beliefs, opinions, expectations, hypotheses, theories, and schemas, which are often used in everyday life to selectively interpret stimuli, do not remain at the cognitive level, since such interpretation permeates actions. Therefore, such cognitive structures expressed through verbal language give meaning to the world [7].

Since semantic networks allow for the conceptualization of knowledge through experience, information acquisition, and the development of skills and abilities [10], the use of the NSN model is justified to apprehend what healthcare professionals understand the neurorehabilitation concept.

To our best knowledge, there is no previous research that has used NSN to study the neurorehabilitation term. Therefore, the aim of this study was to describe the conceptualization of the neurorehabilitation concept by physiotherapists specializing in this field using NSN. Secondarily, the NSN was also studied considering the academic level and years of professional experience in physiotherapy.

## 2. Materials and Methods

A qualitative phenomenological study is presented. The participants were physiotherapists who graduated from the Universidad Rey Juan Carlos, Universidad Complutense and Universidad Europea de Madrid (Madrid, Spain), working in the field of neurorehabilitation. This study obtained favorable approval from the Research Ethics Committee of the Universidad Rey Juan Carlos (reference number: 2802202206122).

The inclusion criteria were as follows: being of legal age, having a degree in physiotherapy and currently working in the field of neurorehabilitation. The exclusion criteria were as follows: non-Spanish nationality, having obtained physiotherapy training in a country other than Spain, not working as a physiotherapist, and failure to adequately complete or provide details for the questionnaires.

A questionnaire was administered through Google Forms electronically and was disseminated via social networks and official channels of the institutions associated with the researchers of this investigation. The instrument used was constructed following the technique of NSN and comprised two sections in which the subjects were required to indicate the most relevant defining word. Firstly, they were asked to define the concept by stating verbs, adverbs, adjectives, nouns, pronouns, without using articles or prepositions [12]. The statement of the first section was as follows: “Next, we invite you to reflect on which words best define the concept of ‘neurorehabilitation’ using verbs, adverbs, adjectives, nouns, pronouns, without using articles or prepositions, nor the word ‘science’. In the box below, indicate at least five words that define the concept”. Secondly, the participants were asked to rank the level of relevance for each defining word mentioned in the previous section. The statement of the second section was as follows: “Then, in the box below, rank the level of relevance that you assign to the words you selected and wrote in the previous box”.

Prior to the data cleansing process, the synonymous relationships between words were analyzed using a double-entry matrix. It is recommended to perform this procedure to condense the information obtained from the original technique of NSN and to avoid losing important information that may be representative in terms of word frequency and hierarchical relevance [13]. Subsequently, a descriptive table was constructed with all the defining words mentioned by the participants, and the main values of the semantic network were calculated based on the proposal by Figueroa, González, and Solís [12] and Valdez [13]. The values considered in the semantic network are as follows [13]: (a) Value J represents the total number of defining words mentioned by the participating subjects. This value is an indicator of the semantic richness of the network, as a greater number of words indicates greater richness. This value was obtained by summing the total number of defining words attributed by the participating subjects; (b) Value M is related to the hierarchy assigned by the subjects to each defining word in terms of semantic weight. To obtain this value, the frequency of occurrence of the defining words was multiplied by the hierarchy assigned to each of them. The set of M values is referred to as the Semantic Association Memory (SAM) set, which constitutes the central core of the semantic network, representing the center of meanings that contribute to the stimulus concept. Typically, it consists of the ten defining words with the highest M value in the network that emerges during the data collection process; thus, the defining words with the highest semantic weight were accounted for.

The Semantic Distance Value (FMG) value refers to the semantic distance between different defining words that have formed the SAM set and was used in all analytical contrasts for this study. This indicator denotes the variability of the different words within a specific set with respect to their central word. It is a measure of distribution that demonstrates the extension emerging in the network of a given set. Thus, words that are closer to the central word have a similarity in semantic importance to the network, while those that approach a percentage value of 0 and move away from the 100% represent a difference from the central word. The analysis is performed by taking as a starting point the defining words with the highest M value in the network, representing 100% in terms of percentage. For each defining word, the M FMG value was calculated by multiplying its M value by 100 and dividing it by the highest M value in the network being analyzed [13].

Additionally, sociodemographic data (age and gender) were registered. To further explore the data, we analyzed the role of academic level and years of professional experience. This study had an exploratory-descriptive scope and was supported by a mixed-methods combination, utilizing qualitative analysis as the primary approach [14]. The single case study approach proposed by Stake [15] was employed to understand the perceptions of Spanish physiotherapists. Some of the analyses were conducted using the SPSS (IBM Corp. Released 2012. IBM SPSS Statistics for Windows, Version 25.0., Armonk, NY, USA) statistical package, and pivot tables were used in Excel 2021 (Version 18.0).

## 3. Results

A total of 191 physiotherapists participated in the present study. Of the total participants, 27.7% (*n* = 53) were male, and 72.3% (*n* = 138) were female. Furthermore, it was observed that 7.3% (*n* = 14) reported having a degree in physiotherapy, 14.7% (*n* = 28) reported having a specialization degree, 59.2% (*n* = 113) had a master’s degree, and 18.8% (*n* = 36) held a doctorate degree. The participants’ areas of work were also assessed, of which 57.1% (*n* = 109) were primarily dedicated to clinical practice, while 42.3% (*n* = 81) combined clinical practice with teaching, research, and management. Finally, among the total sample, 68.6% (*n* = 131) reported having 1 to 10 years of professional experience, 25.7% (*n* = 131) had 11 to 21 years of experience, and 4.7% (*n* = 9) reported having 22 to 34 years of professional experience.

Initially, the defining words that emerged from the network of meanings provided by all the participants were examined. By calculating the value J, which represents the total number of defining words mentioned by the participants, a total of 1247 words were obtained. This quantity denoted a broad semantic richness field regarding the concepts associated with the word ‘Neurorehabilitation’, which are directly related to clinical issues and interventions. Subsequently, the total number of words was refined based on their hierarchical importance and frequency within the sample. Figure 1 displays the diversity of concepts associated with the stimulus provided in the data collection process, identifying specific defining words that carry a stronger semantic weight for the term ‘Neurorehabilitation’.

As highlighted in Figure 1, from the total number of obtained words it is possible to identify several defining words that carry a stronger semantic weight for the neurorehabilitation concept. The general semantic core was primarily composed of the word ‘treatment’ (SAM = 486), followed by ‘recovery’ (SAM = 482), ‘functionality’ (SAM = 479), ‘neuroplasticity’ (SAM = 471), ‘learning’ (SAM = 461), and finally, ‘adaptation’ (SAM = 320). Accompanying this set are the words ‘movement’ (SAM = 210), ‘improvement’ (SAM = 153), ‘independence’ (SAM = 148), and ‘motor control’ (SAM = 148) with lower frequencies than the first ones mentioned. Table 1 provides a detailed frequency analysis of each defining word in the general semantic network, as well as the frequency of the distance between one word and another.

In terms of the frequency detail and FMG between the different defining words, a slight distance is observed between the initial words and the central term ‘treatment’, while the word ‘adaptation’ starts to significantly deviate from it, representing a distance of 65.8%. Subsequently, the words ‘independence’ and ‘motor control’ further separate with a distance of 30.4%. These semantic distances can be visualized in Figure 2.

Next, the semantic weights of the defining words reported by the participants are reported, subdivided by years of experience. Based on this, we have three groups: 1–10 years of experience, 11–21 years of experience, and 22–34 years of experience. While the response frequencies vary among the groups due to the number of subjects involved in each group, it is possible to elucidate the semantic weight that certain defining words had for each group, as well as the relevance attributed to them. For the first group, the defining word with the highest semantic weight was ‘neuroplasticity’ (SAM = 348), followed by the word ‘treatment’ (SAM = 290). For the second group, the defining word with the highest semantic weight was ‘recovery’ (SAM = 197), followed by ‘functionality’ (SAM = 174). In contrast, in the last group, the central word was ‘treatment’ (SAM = 78), followed by ‘learning’ (SAM = 40). The relevance that each group assigns to these defining words reflects a particular emphasis regarding the concept of neurorehabilitation based on the subgroups’ years of professional experience and how their understanding of the term evolves. See Table 2.

The distance between the defining words varied across each subgroup. More specifically, the group with 1 to 10 years of experience showed a slight distance between the first defining words and the central word ‘neuroplasticity’, ranging from a close distance of 83.3% (FMG) with the word ‘treatment’ to 81% with the words ‘learning’ and ‘functionality’. There was then an increase in distance with the word ‘treatment’, with values distributed towards the center or closer to zero, and more pronounced distances with words such as ‘adaptation’, ‘movement’, ‘improvement’, ‘motor control’, and ‘independence’, ranging from 58% for the first of these words to values closer to 40% and 30% for the rest of them. See Figure 3.

In the case of the subgroup with 11 to 21 years of experience, a slight distance was also observed from the central word ‘recovery’ to the second defining word, which is ‘functionality’ (FMG = 88.3%). The tendency to distance themselves from the central word became more evident starting from the word ‘adaptation’ with a distance of 52%, reaching values that varied between 30% with the word ‘rehabilitation’ and the word ‘autonomy’ approaching a value of 0 at 19%. See Figure 4.

The third subgroup, ranging from 21 to 34 years of experience, exhibited a substantial distance between the first defining word, ‘treatment’ (FMG = 100%), showing a significant decrease of 51.2% with the word ‘learning’. Furthermore, distances ranged from 17.9% with the word ‘adaptation’ to values closer to 0 with the word ‘empowerment’ at 12.8%, and with the words ‘brain injury’, ‘normalization’, ‘neurological evolution’, and ‘teamwork’ at 11.5%. See Figure 5.

A third data contrast was conducted to elucidate the differences between the defining words based on the reported academic degrees of the participants. While the response frequencies also varied among the subgroups, certain particularities were identified for each of them. It is worth noting that there was some overlap in defining words that resembled the general SAM set. However, for the group with a degree in physiotherapy, other words were introduced such as ‘nervous system’ (SAM = 25) and ‘empathy’ (SAM = 18), although their central word was ‘neuroplasticity’ (SAM = 51). The remaining groups showed similarities in certain defining words, but differences were observed in the semantic weight assigned to each of them. Table 3 presents that, for the master’s degree and specialization degree groups, the defining word with the highest semantic weight was ‘functionality’ with SAM = 313 and SAM = 98, respectively. Conversely, for the doctorate degree group, the word with the highest weight was ‘treatment’ (SAM = 486). See Table 3.

Differences in distances between the central defining words and their variations can be observed in each group. In the case of the Degree group, the central word was ‘neuroplasticity’, and it showed a slight distance ranging from 80.3% with the word ‘treatment’ to 70.5% with the word ‘recovery’. The trend of distancing from the central word became more pronounced with the word ‘nervous system’ at 49% and the word ‘patient’ at 45%. Subsequently, values closer to 0 were observed with percentage distances ranging from 35% to 25% with words such as ‘empathy’ in the first place, followed by ‘modulation’, and then ‘motivation’, ‘movement’, and ‘independence’. It is worth noting that this group demonstrated a tendency to mention concepts associated with a biological-functional dimension first, followed by concepts more oriented towards an affective dimension. See Figure 6.

For the master degree group, the central defining word was ‘functionality,’ which showed a slight distance from words such as ‘neuroplasticity,’ ‘recovery,’ and ‘learning,’ which were around 80%. The distance became more evident starting from the word ‘adaptation’ at 67%, and then there was a more abrupt tendency towards 0, ranging from 44% with the word ‘movement’ to words like ‘motor control’ and ‘quality of life,’ which were around 25% to 26%, respectively. These latter words alluded to a biological dimension but were not solely limited to their biological nature. See Figure 7.

For the doctorate degree group, a series of defining words were grouped that represent a biological-functional dimension and showed proximity to the central word ‘treatment’. The words ‘recovery’, ‘functionality’, ‘neuroplasticity’, and ‘learning’ had a distance close to 90% from the central word. A greater distance was observed from the word ‘adaptation’ at 65.8%, gradually approaching the value of 0, with distances ranging around 20% and 30% with words such as ‘motor control’, ‘exercise’, and ‘motivation’. See Figure 8.

For the specialization degree group, the central defining word was ‘functionality’ and showed a slight distance from the word ‘recovery’ (FMG = 87.7%). Three defining words were grouped, ranging from a distance of 75.5% with the word ‘learning’, 66.3% with the word ‘treatment’, and 57.1% with the word ‘neuroplasticity’. The distance became more abrupt with the word ‘movement’ at 43.8%, and then increased further with the words ‘perseverance’, ‘improvement’, ‘motivation’, and ‘exercise’, with variations ranging from 23.4% for the first word to 19.3% for the last two. See Figure 9.

## 4. Discussion

This study contributes to research on the concept of neurorehabilitation from the perspective of Spanish physiotherapists using NSN. Additionally, an analysis was conducted considering years of professional experience and academic level, which showed variations in the concept of neurorehabilitation based on these factors. To our knowledge, there is no previous study that has addressed this topic. Khan et al. [16] considered the term neurorehabilitation as a multidisciplinary and interdisciplinary approach that integrates various professional areas in the field of disability. Among these professions, physiotherapy stands out and aims to identify and maximize the potential for quality of life and movement in the realms of promotion, prevention, treatment, habilitation, and rehabilitation [17]. Therefore, in the field of neurorehabilitation, physiotherapists are part of an interdisciplinary team that attends to patients from the early stages, playing a key role in their recovery [18]. Hence, this study aimed to analyze through the NSN design the concept of neurorehabilitation by this health professional group.

A sample of 191 physiotherapists was obtained, with over 72% being women, which can be considered representative of a highly feminized profession, as evidenced by the fact that in Spain, 62.42% of registered physiotherapists were women in 2021 [19]. Furthermore, 59.2% of the participants had completed a master’s degree, and 14.7% had a specialization degree, which is common in the increasingly specialized field of neurological physiotherapy. Additionally, 18.8% of the physiotherapists held a doctoral degree, the highest academic qualification, which is becoming more common among university professors, researchers, and clinicians. Although specialization among physiotherapists is perceived as a necessity, there are numerous paths for physiotherapists to increase their training and improve the quality of their care [20]. It is worth noting that 42% of the participants combined clinical practice with research, teaching, and/or management, despite the fact that 68% had less than 10 years of experience, and 25% had between 11 and 20 years. Considering the research design, using online questionnaires and the channels of dissemination primarily from participating universities, these aspects might have influenced our results.

Among the findings, it is relevant to highlight certain meanings associated with the concept of neurorehabilitation in our sample, which undoubtedly provide a significant contribution to the definition of this term in the literature. Rather than focusing on aspects related to neurological injury or disease, as proposed by the WHO [3], dimensions linked to therapy are emphasized, with words such as ‘treatment’ and ‘recovery’. A recent study [21] that conducted a bibliographic analysis of scientific citation networks on new technologies in the field of neurorehabilitation in the English-language literature seems to align with this trend, with ‘rehabilitation’ and ‘neurorehabilitation’ being the most frequent keywords, while terms like ‘stroke’, ‘spinal cord injury’, ‘cerebral palsy’, and ‘brain injury’ appeared with much lower frequency.

In the present study, other terms emerged as descriptors in the semantic network, such as ‘neuroplasticity’, ‘recovery’, ‘learning’, and ‘adaptation’. There are no previous studies with which to contrast our findings. However, a systematic review published in Colombia in 2015 [22] on physiotherapy in neurorehabilitation found that terms and concepts such as neurofacilitation, motor control, motor learning, and motor patterns, and fundamental principles of neuroscience such as neuroplasticity, neuromodulation, and neurorestoration were closely linked to physiotherapy in neurorehabilitation. It is worth noting that the cited study did not employ the NSN design. However, this is consistent with the idea that one of the main characteristics of neurorehabilitation, compared to rehabilitation, is that it goes beyond the purely physical aspects of the disease, aiming to address the psychological implications of disability and the social environment in which the affected person operates [5].

To our knowledge, there are no previous studies with the same methodological design that would allow us to compare our findings. However, other previous studies have employed NSN in the field of Health Sciences. For example, studies have focused on the concepts of ‘man’ and ‘woman’ for medical students [23], the concept of teaching among medical degree professors [10], the semantic representation of the term ‘medicine’ [8], and the conceptualization of the term ‘human anatomy’ throughout the medical degree program for students [4]. NSN has also been used to investigate the semantic representation of the term ‘medical psychology’ among medical students [9] and to explore the psychological meaning of depression in physicians and psychologists [24], revealing differences in conceptualization based on the participants’ age, educational level, and professional background.

In the present study, it is important to note that the words and concepts used by the participants varied according to their years of experience (Figure 3, Figure 4 and Figure 5). Physiotherapists with more experience more frequently included terms associated with therapy, such as ‘treatment’ and ‘learning’, while for those with less experience, the concepts of ‘neuroplasticity’ and ‘treatment’ were more prominent. It is worth mentioning that 4.7% of the participants had between 22 and 34 years of professional experience. However, in addition to years of experience, it is possible that the increasing trend of specialized treatment for neurological patients could have influenced these results. This is reflected in the studied sample, where 14.7% (*n* = 28) reported having a specialization degree, 59.2% (*n* = 113) had a master’s degree, and 18.8% (*n* = 36) held a doctoral degree. Similarly, Agudelo et al. [22] reported that in recent years, theoretical frameworks such as neurocognition, motor behavior, and fundamental concepts of the International Classification of Functioning, Disability, and Health (ICF) have gained importance in the field of neurorehabilitation. This is consistent with the significant presence of the term ‘functionality’ in the results of the present study. A study on NSN of the concept of ‘subjective well-being’ [25] identified common terms regardless of the participants’ age, gender, or socioeconomic level, although variations in the hierarchy of terms were observed, which is also evident in our study, related to years of professional experience.

Finally, this therapeutic and neuroscientific perspective of neurorehabilitation in relation to motor learning seems to be associated not only with greater professional experience but also with a higher level of specialization. Differences were found based on the participants’ academic level, as those with lower academic levels more frequently referred to meanings associated with a biological dimension, while those with higher academic levels used words associated with the possibility of ‘recovery’ and ‘adaptation’ of patients, which may be more related to an environmental–ecological dimension (Figure 6, Figure 7, Figure 8 and Figure 9). These results could be influenced by the high level of education in the studied sample, with over 18% having a doctoral degree, and the significant dedication to research and teaching, with over 50% of the participants exclusively involved in the clinical field.

This study presents several clinical implications. It was interesting to have better knowledge about physiotherapists define the term ‘neurorehabilitation’ itself, and also to hypothesize why they might select a rehabilitation approach or technique among others, considering a perspective more related to the person (i.e., explained by motor control theories based in the information processing concepts) or those approaches more related to the environment (i.e., explained by motor control theories based in the action and the environment concepts) [26], explained by the terms that they unconsciously or consciously link to the ‘neurorehabilitation’ term.

On the other hand, our results showed a different interpretation of the ‘neurorehabilitation’ in terms of levels of experience and degrees obtained. Physiotherapists with less experience associated the concept of neurorehabilitation with terms related to therapeutic and biological dimensions, while physiotherapists with more experience associated the term with environmental–ecological dimensions. Finally, the higher the academic degree of the physiotherapists, the more they employed concepts linked to an environmental–ecological dimension, while those with lower academic levels more frequently referred to meanings associated with a biological dimension. These findings could explain why physiotherapists could select or vary the approaches or techniques that they apply to their patients, considering their levels of experience and degrees obtained over time [27].

This study has several limitations. In terms of external validity, the sample size and its restriction to physiotherapists graduated from three institutions in Spain limit the generalizability of the findings to all healthcare professionals involved in neurorehabilitation or professionals from other countries. In future studies, expanding the recruitment to other regions and conducting the study in other languages could provide a broader perspective of the concept under study, facilitating comparative analyses. Additionally, including all professionals involved in neurorehabilitation processes would provide a more comprehensive analysis. Finally, while the virtual collection of information using a structured method may limit internal validity, it could enable large-scale research to be conducted.

## 5. Conclusions

The Spanish physiotherapists interviewed used a total of 1247 defining words for the concept of neurorehabilitation. The semantic core of the concept was mainly formed by the words ‘treatment’, ‘recovery’, ‘functionality’, ‘neuroplasticity’, and ‘learning’, which had significant importance.

While physiotherapists with less experience associated the concept of neurorehabilitation with terms such as ‘neuroplasticity’, ‘treatment’, and ‘learning’, which may be more related to therapeutic and biological dimensions, physiotherapists with more experience more frequently used the terms ‘treatment’, ‘learning’ and ‘functionality’ which are associated with an environmental–ecological dimension. The higher the academic degree of the physiotherapists, the more they employed concepts linked to an environmental–ecological dimension, while those with lower academic levels more frequently referred to meanings associated with a biological dimension.

This indicates a variation in the representation that physiotherapists may have of the concept of neurorehabilitation based on their years of experience and academic background. The emphasis placed on one or another significant dimension, be it biological–functional or environmental-ecological, is of utmost relevance as it reflects the semantic richness of the concept of neurorehabilitation. Therefore, the semantic network observed in this study allows us to elucidate the polysemy of the concept of neurorehabilitation, which is composed not only of certain associated words but also the meanings they imply.

## Figures and Tables

**Figure 1 behavsci-13-00972-f001:**
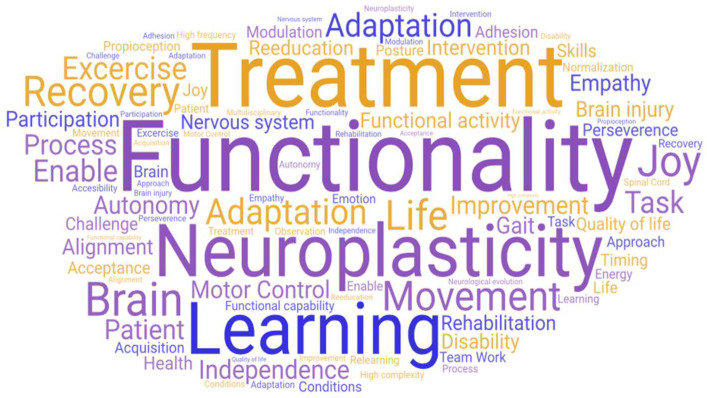
Refined visual representation of the J value.

**Figure 2 behavsci-13-00972-f002:**
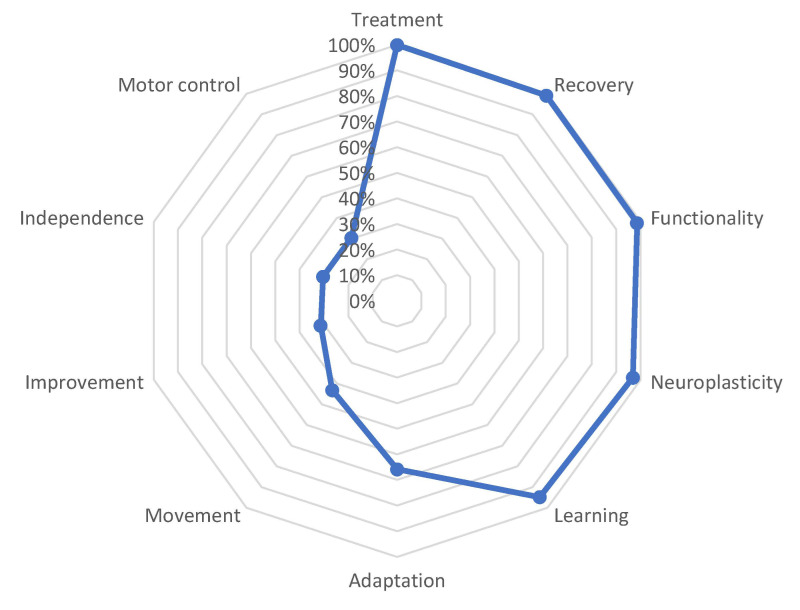
Semantic distance between the different defining words that have formed the SAM set.

**Figure 3 behavsci-13-00972-f003:**
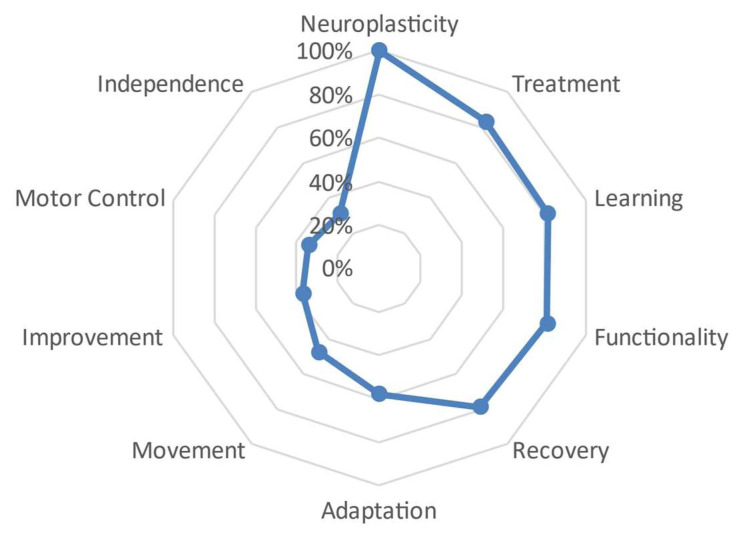
Semantic distance between the different defining words that have formed the set of 1 to 10 years of experience.

**Figure 4 behavsci-13-00972-f004:**
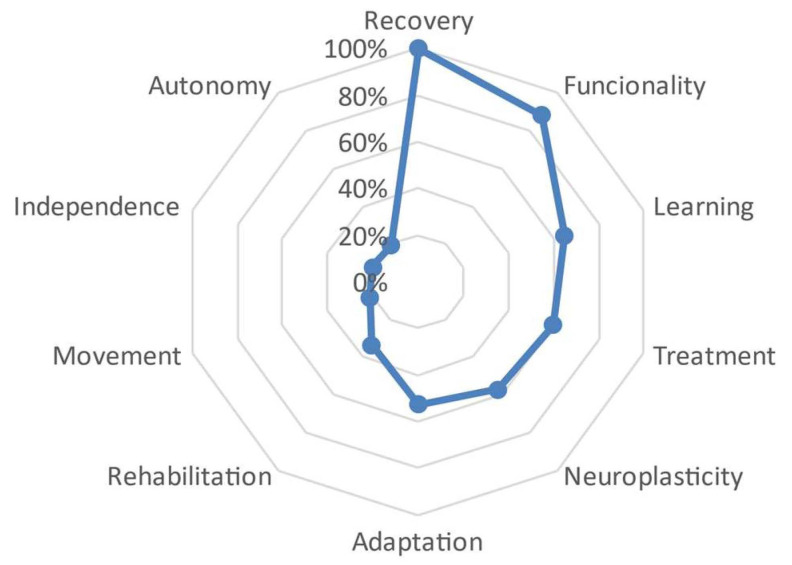
Semantic distance between the different defining words that have formed the set of 11 to 21 years of experience.

**Figure 5 behavsci-13-00972-f005:**
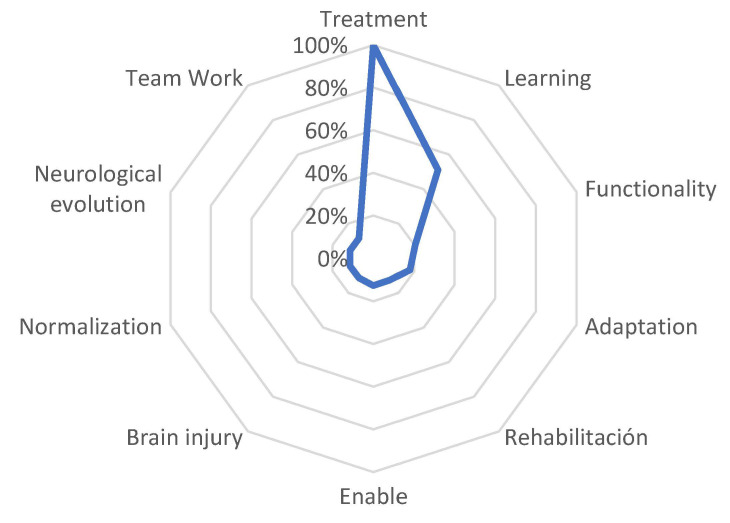
Semantic distance between the different defining words that have formed the set of 21 to 34 years of experience.

**Figure 6 behavsci-13-00972-f006:**
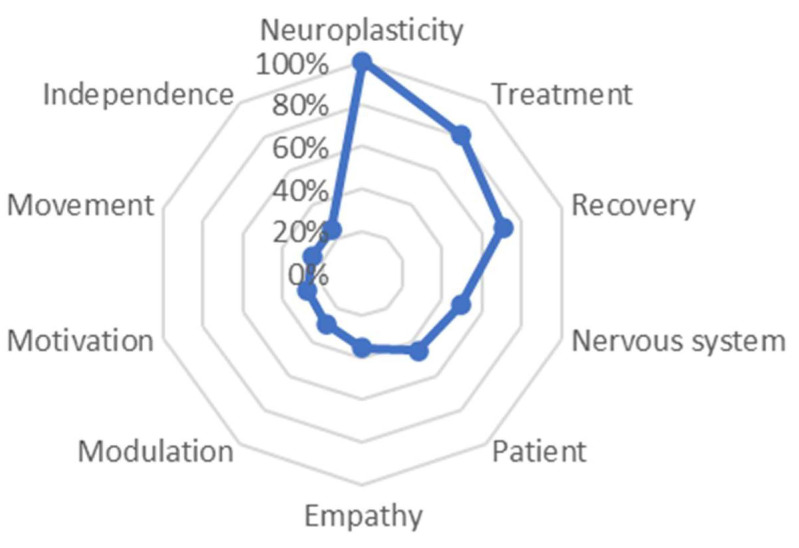
Semantic distance between the different defining words that have formed the degree in physiotherapy group.

**Figure 7 behavsci-13-00972-f007:**
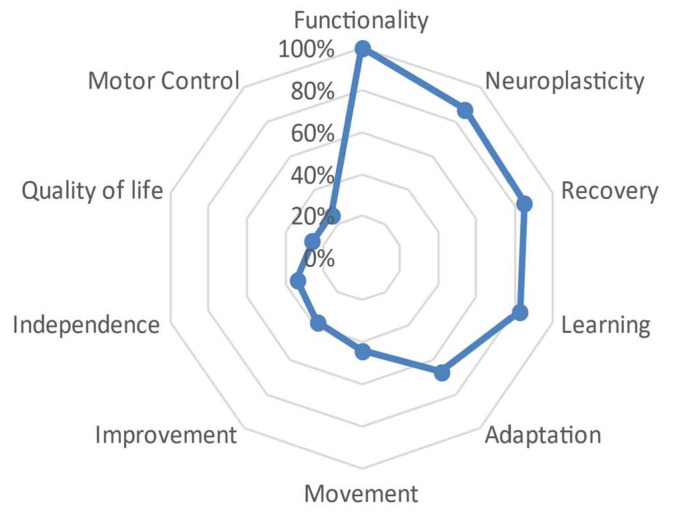
Semantic distance between the different defining words that have formed the master’s group.

**Figure 8 behavsci-13-00972-f008:**
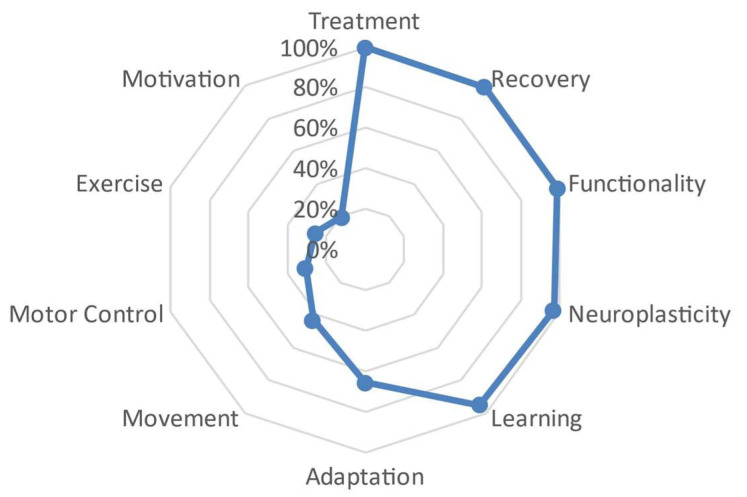
Semantic distance between the different defining words that have formed the doctorate group.

**Figure 9 behavsci-13-00972-f009:**
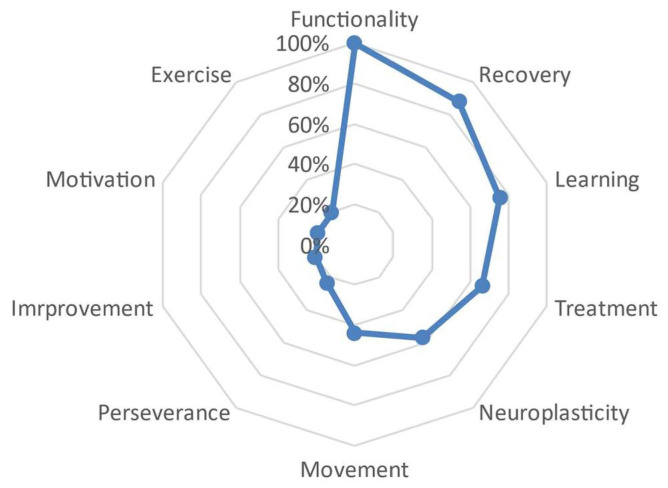
Semantic distance between the different defining words that have formed the specialization degree group.

**Table 1 behavsci-13-00972-t001:** Description of the semantic network corresponding to the overall SAM set.

	M	FREC	FMG
Treatment	486	61	100%
2. Recovery	482	58	99.1%
3. Functionality	479	63	98.5%
4. Neuroplasticity	471	63	96.9%
5. Learning	461	60	94.8%
6. Adaptation	320	42	65.8%
7. Movement	210	32	43.2%
8. Improvement	153	21	31.4%
9. Independence	148	23	30.4%
10. Motor Control	148	20	30.4%

M: semantic weight; Frec: frequency; FMG: semantic distance value.

**Table 2 behavsci-13-00972-t002:** Description of semantic network corresponding to the SAM set according to the years of experience of the sample.

	1 to 10 Years			11 to 21 Years			22 to 34 Years		
	M	Frec	FMG	M	Frec	FMG	M	Frec	FMG
Neuroplasticity	348	43	100%	113	19	57.3%			
Treatment	290	36	83.3%	118	16	59.8%	78	9	100%
Learning	284	38	81.6%	128	15	64.9%	40	6	51.2%
Functionality	283	37	81.3%	174	23	88.3%	26	3	20.5%
Recovery	275	34	79.0%	197	23	100%			
Adaptation	202	26	58.0%	104	14	52.7%	14	2	17.9%
Movement	165	24	47.4%	42	7	21.3%			
Improvement	129	17	37.0%						
Motor Control	120	16	34.4%						
Independence	108	16	31.0%	40	7	20.3%			
Rehabilitation				66	7	33.5%	10	1	12.8%
Autonomy				38	6	19.2%			
Enable							10	1	12.8%
Brain Injury							9	1	11.5%
Team Work							9	1	11.5%
Normalization							9	1	11.5%
Neurological Evolution							9	1	11.5%

M: semantic weight; Frec: frequency; FMG: semantic distance value.

**Table 3 behavsci-13-00972-t003:** Description of semantic network corresponding to the SAM set, according to the academic level of the sample.

	Degree in Physiotherapy			Other Master’s			Other Doctorate			Specialization Degree		
	M	Frec	FMG	M	Frec	FMG	M	Frec	FMG	M	Frec	FMG
Neuroplasticity	51	6	100%	274	35	87.5%	471	63	96.9%	56	7	57.1%
Functionality				313	43	100%	479	63	98.5%	98	11	100%
Learning				259	35	82.7%	461	60	94.8%	74	9	66.3%
Treatment	41	5	80.3%				486	61	100%	65	8	66.3%
Recovery	36	4	70.5%	266	32	84.9%	482	58	99.1%	86	11	87.7%
Independence	13	2	25.4%	106	16	33.8%						
Adaptation				210	26	67.0%	320	42	65.8%			
Nervous system	25	3	49.0%									
Empathy	18	2	35.2%									
Modulation	15	2	29.4%									
Motivation	14	2	27.4%				96	18	19.7%	19	4	19.3%
Movement	13	2	25.4%	138	19	44.0%	210	32	43.2%	43	7	43.8%
Patient	23	3	45.0%									
Improvement				118	17	37.6%				20	2	20.4%
Quality of Life				82	12	26.1%						
Motor Control				79	11	25.2%	148	20	30.4%			
Exercise							125	20	25.7%	19	3	19.3%
Perseverance										23	3	23.4%

M: semantic weight; Frec: frequency; FMG: semantic distance value.

## Data Availability

Data are contained within the article.

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
