# Peer review of "Natural Semantic Networks of the Neurorehabilitation Concept by Spanish Physiotherapists—A Qualitative Phenomenological Representational Study"

_behavsci, 2023, doi:10.3390/bs13120972_

Round 1

Reviewer 1 Report

Comments and Suggestions for Authors

1.           NSN is a way of representing the knowledge of a group of persons.  It relates to a particular topic and the comparison of NSN for different groups provides good insights about the understanding and knowledge of different groups and how it all compares with other groups.  The proposed work is good in presenting this type of comparison in the context of physiotherapists.

2.           The corresponding long forms should accompany all the ‘first usages’ of abbreviations (in the abstract and the remaining manuscript).  At the same location, the words of the long form should be suitably written in Title Case.  Either the style of ‘long form followed by the abbreviation’ (preferably) or the ‘abbreviation followed by the long form’ should be consistently used throughout the manuscript.  After the abbreviation has been defined in the first instance, the subsequent text of the manuscript should not unnecessarily mention the abbreviation and long-form again, and rather, only the abbreviation should be used.

3.           All redundant in-figure titles must be removed, as the figures already have the necessary titles below them.

4.           It is recommended that all occurrences of the usage of double quotes (e.g., for emphasizing/highlighting words/terms/phrases) be replaced with single quotes.  This should not be done for exceptional cases e.g. when verbatim text is reproduced with suitable citations.

5.           There is no difference between Reference No. 6 and Reference No. 23.

6.           Why have the authors used multiple periods in the captions of Fig. 8 and Fig. 9?

7.           Instead of saying, ‘…highlighted in the figure, from..’, it is recommended that the authors make use of the specific Fig. number.  This will improve the readability of the paper, as well as make it more comprehensible.

8.           Do the authors intend to present the same feature of information through Fig. 2 and Fig. 3?  If so, it is recommended that consistent graph formats be used.

9.           Though the authors mention that there are no exactly similar research works, which is appreciable in fact as it highlights the novelty of the proposed work, it will be further appreciable if the authors could present more details (on the existing approaches, domains, results) from whatever existing research works are there.  This will provide the reader with a comprehensive understanding of the proposed concept.  It will also present the proposed work in light of the existing works and hence, improve its presentation.

10.       Overall, I am satisfied with the proposed work, both in terms of manuscript organization, as well as in terms of contributions to the scientific community. However, above corrections are recommended for further uplifting the paper quality.

Comments on the Quality of English Language

None major.

Reviewer 2 Report

Comments and Suggestions for Authors

Title: Natural Semantic Networks of the Neurorehabilitation Concept by Spanish Physiotherapists. A qualitative phenomelogical representational study

Overall assessment:

This study is a semantic analysis of the concept of neurorehabilitation, as conceptualized by professionals in the field with various years of experience. The method used was a questionnaire. Some differences emerged in the use of words to define the concept depending on age group.

While the research question could be of interest, the rationale of this study is not sufficiently described and implications are not shown. Why is it interesting and useful to know how groups of professionals without varying levels of experience and degrees obtained will define neurorehabilitation? What does it mean in the clinic? Why are there generational/experiential trends? What elements in the training, in the progress of science etc. can explain these findings? These elements need to be discussed in the discussion section. That section also contains a lot of redundant information, or information that needs to be placed in introduction. Therefore, the manuscript needs significant improvement before this work can be published.

Main comments:

Abstract

The sentence "Physiotherapists with less experience associated the concept of neurorehabilitation with the terms "neuroplasticity," "treatment," and "learning." is not very informative because it is not giving any point of comparison in the data. compared to what other group? What words did they give?

1.     Introduction

What might be alternate ways to look into the meanings attributed to neurorehabilitation besides NSN? A presentation of other methods if they exist and models, and how well they do in comparison to each other would help situate the choice of using NSN.

What are some of the conceptual differences that healthcare professionals have of the concept (again to situate the problem)?

What is FMG? This needs to be spelled out at the first occurrence.

2.     Materials and Methods

Some description of what is the technique of NSN to build the questionnaire would be helpful.

The foreshadowing on l. 154 could be grouped with all the analyses done in he present study. As it stands, the reader is left not knowing exactly what was done in terms of comparing years of experience and academic level – e.g. between who, and why does it matter?

I think the "rule of three" concept is only known as such in France and Spain – I would just remove reference to it and say "The analysis is performed by taking…".

Shouldn't this sentence: "For each defining word, the M FMG value was calculated by multiplying its M value by 100 […]"?

3.     Results

Figure 2: It may be more visually impactful and logically aligned to reverse the y-axis such that Treatment is at 0%: it's the core word so it doesn't deviate at all from itself. Then show how the next words deviate in percentage difference, e.g. Recovery would deviate by 100-99.1 = .9% etc. EDIT AFTER SEEING NEXT FIGURES: Why not graph Figure 2 the same way as Figures 3-5?

The caption of Table 2 needs to mention that it's showing data in terms of years of experience. Same comment for Table 3, the caption can be more descriptive so that someone who does not read the paragraph above can still understand the data in the table, e.g. the column "Degree" could be replaced by "Degree in Physiotherapy", then "Other Master's", "Other Doctorate". Similarly, Fig. 9 could be titled "Specialization degree group" to be consistent in the use of specific terminology.

4.     Discussion

Any data already presented in the Results section does not need to be repeated in the Discussion (e.g. number of participants, SAMs, percentages…)

Definitions of terms should all be in Introduction. The Discussion section is there to discuss results and expand on them, and only new information should be present.

Paragraphs 1, 2 and 3 should be moved from the discussion and the redundant information can be omitted.

The first sentence of the paragraph starting l. 371 is not discussed. The next sentence is jumping topics without a transition.

Spell out SNA l. 378.

5.     Conclusion

It is said that the different emphases found through this study are of utmost relevance but it is not said for what and how knowing about semantic richness affects the field and practice. This is a key part of the paper that needs to be developed to show the rationale and relevance of the study.

Minor comments:

There is a typo in "phenomelogical" in the title.

This sentence l.99-100 is difficult to understand and has grammaticality issues: "It is easily applicable due to its dynamic nature, as it interacts the contents and structure of the semantic network with other memory processes".

l. 177 Master's degree,

Comments on the Quality of English Language

See above - some minor comments.

Reviewer 3 Report

Comments and Suggestions for Authors

The paper describes the conceptualisation of concept of neurorehabilitation by Spanish physiotherapists specialising in this field using natural semantic networks (NSN). Authors conduct a thorough research on the subject and present main terms forming the semantic core of the concept. Results are clearly presented and well-explained. The paper is well-structured, it is very interesting, and I have learned a lot while reading it.

There are some minor drawbacks that should be considered to improve the quality of the paper before publishing:

1) Please revise abstract, it should be an overview of the results without going into details. Currently abstract contains detailed results mentioning numbers and specific terms associated with the concept of neurorehabilitation, and those do not provide much information without full context that can be found later in the paper.

2) There is no comparison between the used method and any other methods from related research. If there are no studies on the subject it should be clearly stated in the introduction. Lines 331-332 mention that there are no previous studies that has addressed the differences in the concept of neurorehabilitation based on years of professional experience. Mentioning this in the introduction section might increase the quality of the paper and showcase the new knowledge that paper brings in a clearer way.

3) Add paper structure at the end of the introduction section briefly outlining all further sections.

4) Lines 333 – 346. Discussion section begins with key definitions of terms used in the paper (i.e. neurorehabilitation). However, it has already been done in the introduction section. Please revise this without restating the same/similar information.

5) Discussion section does not have any figures. It might be beneficial to combine or use some pictures from results section and show them in this section to clearer show the context.

6) Please do not break words across a line-break by means of a hyphen. It is generally not advisable to do in scientific papers. If a word does not fit on a line, move it fully to the next line.

Round 2

Reviewer 1 Report

Comments and Suggestions for Authors

The suggestions of the previous review round have been well considered by the authors.

Reviewer 2 Report

Comments and Suggestions for Authors

Thank you for your responses to my comments, they helped clarify the study. A few minor revisions are suggested.

Note: In the future, please make sure to update the line numbers of your changes after you have done revisions as the lines shift as a result. I could not find several of the changes listed.

Here are my responses by the numbers provided in the author’s response:

1.     Rationale: the lines indicated do not correspond to yellow highlights. Some of the highlighted additions do explain the rationale better.

2.     Ok

3.     Ok. The word “Undoubtedly” l. 109 makes it sound as if the authors are not sure these other methods exist. If they do exist, then this word can be removed and the sentence start directly with “There are other types…”.

4.     Ok, but I see an equivalent of this paragraph in the discussion rather than in the introduction.

5.     Ok

6.     Ok

7.     Ok – this phrase has a small grammatical error: “in order to gain a better interpretation of the results”. I would replace this however by “To further explore the data, we analyzed the role of academic level and years of professional experience”. With the initial formulation, I would expect that previous work has been done on that topic and that it suggests the need to check for these two variables, but this work is no presented in introduction, suggesting it does not exists, therefore this aspect of the present study is exploratory. If not, then some review of previous work on the topic would have to be included.

8.     Ok

9.     Ok

10.  Ok

11.  Ok

12.  Ok

13.  Ok

14.  Ok

15.  The line numbers do not correspond to the changes.

16.  –

17.  Ok

18.  Ok

19.  Ok

20.  Ok

Comments on the Quality of English Language

Minor edits suggested.
